# Elastomer Nanocomposites: Effect of Filler–Matrix and Filler–Filler Interactions

**DOI:** 10.3390/polym15132900

**Published:** 2023-06-30

**Authors:** Liliane Bokobza

**Affiliations:** Independent Researcher, 194-196 Boulevard Bineau, 92200 Neuilly-Sur-Seine, France; liliane.bokobza@wanadoo.fr

**Keywords:** reinforcement, rubbers, nanoparticles, in situ generated particles, spherical particles, carbon dots, carbon nanotubes, graphenic materials

## Abstract

The reinforcement of elastomers is essential in the rubber industry in order to obtain the properties required for commercial applications. The addition of active fillers in an elastomer usually leads to an improvement in the mechanical properties such as the elastic modulus and the rupture properties. Filled rubbers are also characterized by two specific behaviors related to energy dissipation known as the Payne and the Mullins effects. The Payne effect is related to non-linear viscoelastic behavior of the storage modulus while the Mullins or stress-softening effect is characterized by a lowering in the stress when the vulcanizate is extended a second time. Both effects are shown to strongly depend on the interfacial adhesion and filler dispersion. The basic mechanisms of reinforcement are first discussed in the case of conventional rubber composites filled with carbon black or silica usually present in the host matrix in the form of aggregates and agglomerates. The use of nanoscale fillers with isotropic or anisotropic morphologies is expected to yield much more improvement than that imparted by micron-scale fillers owing to the very large polymer–filler interface. This work reports some results obtained with three types of nanoparticles that can reinforce rubbery matrices: spherical, rod-shaped and layered fillers. Each type of particle is shown to impart to the host medium a specific reinforcement on account of its own structure and geometry. The novelty of this work is to emphasize the particular mechanical behavior of some systems filled with nanospherical particles such as in situ silica-filled poly(dimethylsiloxane) networks that display a strong polymer–filler interface and whose mechanical response is typical of double network elastomers. Additionally, the potential of carbon dots as a reinforcing filler for elastomeric materials is highlighted. Different results are reported on the reinforcement imparted by carbon nanotubes and graphenic materials that is far below their expected capability despite the development of various techniques intended to reduce particle aggregation and improve interfacial bonding with the host matrix.

## 1. Introduction

It has been long established that the incorporation of mineral fillers such as carbon black or silica in elastomeric materials leads to greatly improved properties with regard to the unfilled system which is generally a weak material [1]. But incorporation of large amounts of inorganic fillers is often required to reach the desired properties. The reinforcing properties of conventional fillers such as carbon black and silicas have been widely demonstrated in the literature [2,3] as well as the basic processes contributing to the reinforcement effect [4,5,6,7,8,9,10]. The extent of property improvement has been shown to depend on several parameters such as the volume fraction of filler, the morphology of the particles, their state of dispersion in the host polymeric matrix and essentially their surface characteristics that determine the interface of the polymer–filler system. This interface that plays a crucial role on the properties of the resulting composite can be studied by various techniques able to identify the interacting species in each phase.

Nanoscale fillers with isotropic or anisotropic sheet-like or needle-like morphologies bring much more improved properties when introduced in the host matrix at a very low content on account of their nanoscale dimensions that create a very large polymer–particle interfacial region and of their ability to orientate during processing or mechanical stretching. This interfacial region is of critical importance in nanocomposite properties, and analyzing its characteristics and its role on the macroscopic properties of the material allows the design of composites with specific applications.

Different filler morphologies such as nanospheres, nanotubes or nanoplatelets have been used for the reinforcement of elastomeric materials. Anisotropic fillers with one- or two-dimensional nature are able to orientate during processing or mechanical stretching which has a large effect on the reinforcement of the final material in the direction of alignment. In addition to the basic mechanisms of filler reinforcement discussed below, each type of nanofiller imparts specific reinforcement effects and different properties to the resulting nanocomposite on account of its characteristic structure and geometry. Before reporting some studies using different types of nanofillers in rubber matrices, the present paper briefly recalls the basic mechanisms of reinforcement in conventional rubber composites by taking the option of discussing the mechanical data arising from the author’s research related to two different filled systems that display a strong and a weak polymer–filler interface (silica-filled poly(dimethylsiloxane) and silica-filled styrene–butadiene rubber).

## 2. Basic Mechanisms of Filler Reinforcement in Conventional Composites

The addition of filler particles in a polymeric media is well known to affect the macroscopic mechanical properties of the composites. Specific features of filler reinforcement can be evidenced through basic tensile measurements of elastomeric composites in the form of the nominal stress, σ (force divided by the undeformed area) against strain, ε. Figure 1a compares the tensile properties of an unfilled and silica-filled poly(dimethylsiloxane) (PDMS) networks. The main effect is an increase in the composite stiffness that first involves the inclusion of rigid particles in the soft polymeric medium. This effect depends on the volume fraction and on the aspect ratio of filler according to the Guth equation in the case of filled elastomeric composites [11]. This model, only based on the aspect ratio, f, of the particles and on the volume fraction, ϕ, of filler, can account for the change in modulus at low filler content as in Equation (1):E = E_0_ (1 + 0.67 f ϕ + 1.62 f^2^ ϕ^2^).(1)

Another contribution to the elastic modulus arises from polymer–filler interactions leading to an increase in the effective degree of cross-linking. This contribution can be evaluated by equilibrium swelling and by measurements of chain orientation [12].

At high filler concentrations, the formation of a percolating filler network offers a large contribution to the modulus at low strains [13]. The filler network that results from filler–filler interactions is well known to affect the dynamic viscoelastic behavior of the composite. It is responsible for the typical non-linear viscoelastic behavior of the storage modulus G’ called the “Payne effect” [14] that exhibits a drop with increasing strain amplitude (Figure 1b). This effect is mostly explained by a disruption of the filler agglomerated structures. The Payne effect, which is strongly linked to the state of filler dispersion that depends on the surface characteristics of the particles and on the strength of the polymer–filler interactions, has been widely discussed in the literature [15]. Filler agglomeration is much more important in pyrogenic silica by hydrogen bonding through silanols present on the silica surface. Surface treatment intended to deactivate part of the silanols improves the filler dispersion by decreasing filler–filler interactions and consequently the amplitude of the Payne effect as shown in Figure 1b in the case of silica-filled PDMS networks.

The Payne effect can also be visualized by the Mooney–Rivlin representation [16,17] of the tensile measurements of rubber composites by plotting reduced stress [σ*] defined by the quantity [σ*] = σ/(α − α^−2^), where α is the extension ratio (ratio of the final length of the sample along the direction of stretch to that of the initial length before deformation, ε = α − 1). The strong decrease in stress observed at low deformations for the filled sample is due to the breakdown of the filler network or at least to agglomerated filler structures (Figure 1c) while the strong increase displayed at high deformations is attributed to the limited chain extensibility on account of strain-amplification effects due to the inclusion of undeformable particles. The theoretical model developed by Bueche [18] relates the extension ratio of the chains in the rubber material α_rubber_ to the macroscopic extension ratio α of the composite and to the volume fraction of filler ϕ by the following expression:α_rubber_ = (α − ϕ^1/3^)/(1 − ϕ^1/3^).(2)

The Bueche model considers a unique interparticle distance which means that the strain amplification is uniform within the whole sample. In fact, the strain amplification depends on the local filler content. On account of inhomogeneous strain fields, strain amplification causes more pronounced overstrains of polymer chains in filler-rich and more aggregated areas of the sample [19]. Chains that reach their limit of extensibility break or slip from the particle surface and do not contribute anymore to the modulus causing what is called the Mullins effect or stress-softening effect, characterized by a lowering in the stress when the sample is extended a second time (Figure 1d). Figure 1d displays the stress–strain responses of a silica-filled PDMS sample submitted to four cycles (stretch, release of the strain and stretch again) at various deformations. Equilibrium swelling measurements performed on pre-stretched samples at different strains confirm the loss of network chains during extension of filled rubbers [20,21].

It is of interest to mention that silica-filled PDMS systems display a strong interface created by hydrogen bonding between the silanols and the oxygen atoms of the PDMS chains. A tailored treatment of this interface can be used to optimize the properties of the resulting composite. In the case of poor compatibility between the elastomer chains and the filler surface as in the case of silica-filled hydrocarbon rubbers, coupling agents such as the bis(3 triethoxysilylpropyl) tetrasulfide (TESPT), commonly abbreviated “Si69”, can be used to help the filler dispersion and to improve the adhesion between the organic and inorganic phases. This molecule bears ethoxy groups that can interact with silica and a tetrasulfane function that can react with the hydrocarbon rubber. Figure 2a shows that a large increase in stiffness is obtained when the SBR (styrene–butadiene rubber) is filled with silica compounded with a coupling agent. In the absence of a coupling agent, filler networking dominates at low strains reflected by the strong increase in the initial modulus (Figure 2a), the strong amplitude of the Payne effect (Figure 2b) and the strong decrease in the reduced stress at small deformations (Figure 2c). Figure 2d shows that the Mullins effect is less important without the coupling agent, thus pointing out the role played by polymer–filler interactions and the interface between the two phases in the stress-softening phenomenon. A study carried out on SBR/silica composites by Bernal-Ortega et al. [22] shows that the way of modifying the silica surface strongly affects the properties of the resulting material. The use of pre-modified silica appears as an alternative to the in situ silanization of silica during the mixing process.

The reader can find in the recent work of Robertson and Hardman [10] an analysis of the reinforcement of rubber provided by carbon black (CB) which is certainly the most prevalent filler in the rubber industry, especially in tire technology. Besides a description of the structural and chemical characteristics of CB particles, the authors point out the importance of processing conditions on the reinforcement of elastomers and report various chemical modifications aimed at improving the polymer–filler interactions.

## 3. Nanospherical Particles

As already mentioned, a homogeneous and uniform dispersion of filler particles in the soft matrix has a large influence on the physical performance of the rubber material. In the conventional procedure for composite preparation, filler particles are blended into the polymers before the cross-linking process. The particles tend to agglomerate and the resulting composites are rather inhomogeneous which is detrimental to the mechanical properties since filler agglomerates may lead to crack initiation, crack propagation and failure. In the work of Huneau et al. [23] devoted to the analysis of the fatigue crack initiation in a carbon black-filled natural rubber, it is shown that the crack initiation mechanism around CB agglomerates involves a debonding at the polymer–filler interface, thus reflecting the strong internal cohesion of the agglomerates. Gehling et al. [24] very recently provided a general overview on the fatigue behavior of elastomeric components and analyzed the key factors able to affect the crack growth behavior. Besides the configuration of the filler particles, their dispersion and distribution in the elastomeric matrix, the authors pointed out the important role of the strain-induced crystallization phenomenon that increases the resistance to crack growth. Cracks can also be generated by residual stresses arising during the composite processing. These residual stresses that can be detrimental to the performance of the resulting composite can be attributed to the differences in thermal expansion coefficient between matrix and filler [25,26,27,28].

As discussed previously, the incorporation of conventional silica to hydrocarbon rubbers leads to the formation of a strong filler–filler network due to a weak interfacial adhesion between the two phases. In addition, in sulfur-cured systems, silica reacts with the chemical ingredients of formulation which lowers the overall cure state. The use of silane coupling agents in combination with silica in nonpolar polymers decreases the filler–filler interactions and simultaneously increases the polymer–filler interactions.

Fillers can be generated in the elastomeric matrix by a sol–gel process based on the hydrolysis and the condensation of inorganic alkoxides used as precursors such as tetraethoxysilane (TEOS), for example, for silica formation in the presence of a catalyst. This synthetic route leads to almost “ideal” dispersions with small and well-dispersed particles within the host matrix. The morphology of inorganic generated structures depends on the hydrolysis and condensation conditions, especially on the nature of the catalyst used to accelerate the gelation process, and also on the nature and reactivity of the precursor. Different methods can be used to carry out the in situ formation of the filler particles which can be precipitated before or after the polymerization of macromolecular chains and before, after or during the cross-linking process for elastomeric systems.

Mark and coworkers [29,30,31] and references therein widely discussed the in situ reinforcement of PDMS elastomers by silica and titania particles with the examination of various synthetic protocols and the analysis of the dependence of particle size on network chain lengths, filler contents and catalyst concentrations.

Less aggregated filler structures are obtained when the sol–gel process is carried out in the already preformed networks due to a steric limitation for cluster growth by the network chains. Breiner et al. [31] demonstrated that silica particle sizes are the smallest for the smallest values of mesh size of the network because of constraining effects from the network chains. SiO_2_ and TiO_2_ nanoparticles have been successfully generated by swelling the already preformed PDMS networks with tetraethoxysilane (TEOS) and titanium n-butoxide (TNB) as silica and titania precursors, respectively, in the presence of dibutyltin diacetate as catalyst [32,33,34]. The preformed PDMS networks have been obtained by end-linking polymer chains (hydroxyl-terminated PDMS with an average molecular weight of 18,000 g.mol^−1^) by means of a multifunctional crosslinking agent (TEOS) in the presence of stannous-2-ethyl-hexanoate used as a catalyst. Besides better dispersions than those observed by conventional blending of the particles, the two types of particles were seen to display different morphologies. While titania particles are spherical in shape with diameters between 20 and 40 nm in size and almost connected in a branched network structure even at a filler loading as low as 10 phr, at a similar filler loading, smaller silica domains around 5 nm in diameter and rather diffuse polymer–silica interfaces are observed. At higher silica contents, a fine morphology of the silicate structure is still obtained, suggesting an interpenetrated polymer–silica network.

Modulus, tensile strength and elongation at break increase with the amount of the in situ-generated particles but better reinforcement at large elongations is observed with silica particles (Figure 3a). The improved dispersion results in increased interactions with polymer chains that contribute to the formation of an immobilized or glassy polymer layer observed by NMR [32], differential scanning calorimetry and dielectric techniques [35,36,37]. At higher filler loadings, one would assume that strong confinements of polymer chains between filler particles extend the region of reduced mobility outside of the glassy interface layer, thus allowing, at a certain amount of filler, a percolation via an overlap of these constrained regions. This may explain the change in the shape of the stress–strain curve of the silica-filled networks above 30 phr. At 41 phr of silica, a strong increase in stress, the low deformation range is followed by a smaller strain dependence, suggesting a plastic behavior which may be understood by the formation of an interconnecting filler network. The same shape is observed in the stress–strain curves of titania-filled networks even at the lowest filler content (Figure 3a). Thermal and dielectric techniques show that interfacial effects are more pronounced in the titania composites than in the silica composites with a higher polymer fraction of reduced mobility [36,37].

The mechanical behavior of in situ filled systems, different from that of conventional composites, is typical of interpenetrated double networks observed in hydrogel systems and double network elastomers based on a toughening strategy consisting of a covalently cross-linked brittle first network and a stretchable second network [38,39,40]. The interpenetration of two networks of different properties imparts to the resulting material specific features such as yielding and large hysteresis due to the fracture of covalent bonds in the first network. The silica-filled PDMS display a small amount of stress-softening before the formation of the filler network, which may be related to the good filler dispersion and the narrow distribution of chain lengths between filler particles. But significant residual deformation (Figure 3e,f) that becomes much more important above the percolation threshold is observed as the result of the breaking up of both the filler network and the polymer-filler bonds (Figure 3f). It is interesting to mention that no Payne effect is observed in the silica-filled systems even after the formation of the filler network, which probably means that the silica network is resistant to the mechanical work applied within the strain range investigated (Figure 3b). A Payne effect is observed for the titania-filled PDMS networks which is consistent with the tensile results (Figure 3a,c).

The technique of in situ-generated particles in organic rubber matrices considered as an interesting approach for rubber reinforcement [41] has been used for the generation of titania in acrylonitrile butadiene rubber and a solution of styrene–butadiene rubber [42,43]. In both cases, significant improvement in the mechanical properties of the resulting composites is observed despite the poor compatibility between the styrene butadiene rubber and the hydroxyl groups on the titania surface. This confirms the results already obtained in in situ silica-filled natural rubber where the in situ filling process has been conducted in the already preformed networks [44]. The sol–gel approach used to generate zirconia particles into a nitrile rubber has been shown to yield a much higher reinforcing effect than externally filled composites [45]. The use of TESPT as a coupling agent further enhances the mechanical performance of the composites.

In recent years, carbon dots (CDs), a new form of carbon nanomaterials, have emerged as a promising zero-dimensional structure with an exceptionally small size (a few nanometers) and intrinsic photoluminescence properties. Additionally, the presence of oxygen functional groups on their surface able to give rise to a strong interaction with the host medium has drawn a considerable interest towards CDs as reinforcing nanofillers for polymeric matrices [46,47]. CDs can be obtained by breaking carbon structures like carbon, graphite, carbon nanotubes, etc., using arc discharge, laser ablation or electrochemical oxidation (top-down approach). The second synthetic route (bottom-up approach) uses starting materials based on acid reagents such as citric acid, ascorbic acid or, more interestingly, inexpensive “green” sources such as fruits, vegetables, flowers, etc. The reader can find more details in the literature on the synthetic aspects of these fascinating materials as well as on their characteristics and applications [46,47,48,49,50].

Regarding the application of CDs in the field of polymer nanocomposites, Sreenath et al. [51] report results obtained on carboxylated acrylonitrile butadiene (XNBR) latex filled with amine and carboxyl functionalized CDs prepared from citric acid and glycine. 1-(3-dimethylaminopropyl)-3-ethylcarbodiimide hydrochloride and (N-hydroxy-succinimide) have been used as coupling agents to covalently conjugate CDs to XNBR. Tensile strength and modulus are seen to increase with an increase in the CDs content up to 2 phr, and the decrease in the mechanical properties at a 4 phr loading is attributed to an agglomeration of CDs in the matrix (Figure 4). The TEM image of the pristine CDs shows spherical particles with average diameters in the range of 2–4 nm (Figure 4a), while Figure 4b related to the latex with 2 phr of CDs reveals a uniform dispersion of the particles, but an average thickness in the range of 60–70 nm was attributed the formation of clusters from interactions between the functional groups present on the carbon dot surface [52].

Fernandes et al. [53] report an interesting approach that consists in generating in situ carbon dots in a polymer matrix, thus avoiding the use of solvents required to preform the carbon dots before being incorporated into the host matrix. The work relies on the pyrolytic decomposition of ethanolamine within polyethylene, polypropylene and polyethylene glycol. This strategy can open the way to the large-scale production of carbon dot-filled fluorescent polymer nanocomposites. Yu et al. [54] use a CDs slurry (prepared by mixing the CD powder with water) and a compatibilizer (a polybutadiene grafted with maleic anhydride in order to modify the interfaces during the slurry processing) to prepare styrene–butadiene/CD composites. But the size of the CDs in the polymer matrix (50–200 nm) remains much larger than that of the primary CD particles expected to be between 2 and 5 nm. The authors observe the same mechanical performance as that imparted by silica at the same filler loading while one would have expected better reinforcing efficiency of carbon dots. This raises the challenge, as in the case of conventional fillers, to overcome the problem of CDs agglomeration in order exploit the full potential and the superior properties of these new fillers. Jin et al. [55] prepare a fluorescent elastomer based on silicon filled with 30 phr of silica and a CD fluid obtained by mixing the CDs (fabricated from citric acid monohydrate and ethylenediamine) with water. The authors report a fine CD dispersion within the polymer matrix ascribed to the presence of water in the CD fluid and to hydrogen bonding between hydroxyl groups of silica and hydroxyl/carboxyl/amide groups of the CD surface. The low CD loading (less than 1 phr) has no effect on the mechanical performance of the resulting composites but imparts to the matrix a strong fluorescence emission under UV excitation. More interestingly, the fluorescence intensity increases upon mechanical deformation of the composite. Shauloff et al. [56] demonstrate the correlation between the fluorescence intensity of the CD-filled composites and the mechanical measurements. This strain dependence of the fluorescence emission of the CDs embedded in a polymer matrix makes CD fluorescent dyes particularly interesting for tensile sensing and mechano-optical tuning.

## 4. Rod-Shaped Particles

Since their discovery, carbon nanotubes (CNTs) have been the most widely used as a one-dimensional nanofiller on account of their high aspect ratio and their unique mechanical, electrical and thermal properties that make them the ultimate reinforcing filler for advanced composite materials. Their high aspect ratio is expected to yield the percolation threshold at very low filler loadings and impart electrical conduction to insulating media. But the transfer of these exceptional properties to the host matrix requires homogeneous dispersion, an alignment and strong interfacial bonding [57]. In fact, the full realization of their capability is hard to achieve due to the poor interfacial bonding with the polymer matrix, their tendency to bundle together and their possible degradation during processing. The reported results in terms of reinforcement remain far below their theoretical potential, but, nevertheless, substantial improvements in stiffness have been reported upon addition of multiwall carbon nanotubes (MWCNTs) in elastomeric matrices despite the presence of bundles that act as failure points and reduce the expected property improvement. The obtained reinforcing effects have been mainly attributed to the high aspect ratio of the tubes [58,59,60,61,62,63].

In order to achieve the full potential and exploit the superior properties of CNTs, different dispersion processes and various techniques of composite manufacturing are described in the literature in order to reduce particle aggregation and improve the composite properties [64,65,66,67,68,69,70,71]. Ma et al. [64] described various types of mechanical methods (ultrasonication, shear mixing, calendaring, ball milling, stirring and extrusion) as well as physical and chemical functionalization intended to modify the surface properties of the tube. In the review of Rennhofer and Zanghellini [65] dealing with the ultrasonic dispersion process which is one of the most widely used method to break up the agglomerates, damage that may lead to shortening and changes in the surface of CNTs are described as well as the factors that have to be taken into account for optimal dispersion and minimal damage. The influence of CNT shortening on the electrical percolation in a styrene–butadiene-based star block copolymer has been investigated by Staudinger et al. [72].

The physical or non-covalent functionalization include the wrapping of polymer around the tube and the use of surfactants or the endohedral method in which guest atoms or molecules are stored in the inner cavity of CNTs. The non-covalent functionalization does not cause damage to the tube structure while the chemical functionalization based on the introduction of functional groups on the tube surface can degrade the electrical properties due to a disruption of the surface-conjugated π network. Ponnamma et al. [67] provided an overview of various preparation methods of CNT-reinforced elastomers intended to promote and increase the dispersion and stressed the alignment of CNTs in the matrix that can strongly affect the composite properties. In two recent reviews, Speranza [69,70] described the different approaches for the modification of the surface chemistry of carbon nanotubes and also of other carbon nanomaterials. These reviews highlight the different functionalization techniques developed for a desired surface chemistry required for a specific application.

Fukushima et al. [73] showed that an interesting approach to improve the CNT dispersion is the use of room-temperature ionic liquids of imidazolium ions that are seen to display a strong affinity toward the π-electronic surface of single-walled carbon nanotubes and tend to untangle the bundles. Examples of organic cations and anions commonly used to obtain ionic liquids and different uses and applications in combination with CNTs are reported in the paper of Polo-Luque et al. [74]. Das et al. [75] tested a series of ionic liquids in order to identify the molecule that yields better compatibility between diene elastomers and MWCNTs. Their results show that the 1-allyl-3-methyl imidazolium chloride (AMIC) added to the carbon nanotubes provides a strong level of reinforcement to a blend of solution–styrene–butadiene and polybutadiene rubber even at a 3 phr CNT loading. At this low filler concentration, the 100% modulus, tensile strength and elongation at break are significantly improved with regard to the unfilled rubber, and a high conductivity is found compared to that displayed by other composites at a 3 phr CNT loading prepared with other ionic liquids. It is suggested that ionic liquid AMIC may act as a coupling agent since the double bond in its tail could interact with that of the diene rubber molecules and the imidazolium group with the CNT surface. Krainoi et al. [76] used an ionic liquid (IL), namely the 1-butyl 3-methyl imidazolium bis(trifluoromethyl-sulphonyl) imide, for the preparation of natural rubber filled with MWCNTs. It was shown that the addition of IL increases the moduli but lowers the tensile strength and elongation at break with regard to the gum vulcanizate. Moreover, a larger Payne effect is observed as a result of a better dispersion that leads to the formation of a three-dimensional filler network (Figure 5). It is worth noting that an increase in the amplitude of the Payne effect is associated in this case with a good level dispersion of carbon nanotubes, while in conventional composites, an improvement of filler dispersion decreases the Payne effect [61,77].

Galimberti et al. [78] showed that the addition of a small amount of MWCNTS to a composite heavily filled with carbon black promotes a strong Payne effect attributed in another study to the high surface area of the tubes [79]. As mentioned by the authors, the strong dissipation of energy associated with the pronounced Payne effect displayed by CNT-based elastomeric composites prevents the large-scale use of CNTS for dynamic mechanical applications especially in the field of tires where it impacts the fuel consumption. In the work of Bernal-Ortega et al. [80], MWCNTS were surface-modified with oxygen-bearing groups and sulfur in order to decrease the energy dissipation that affects the rolling resistance of tire tread compounds. A reduction in the energy dissipation and in the loss factor at 60 °C is effectively obtained in the functionalized CNTs with regard to the non-modified CNT-filled samples, but better electrical and tensile properties are displayed by pristine fillers. The authors suggested a poor dispersion in the rubber matrix, reduced aspect ratio and a disruption of the π-electron system for the chemically modified nanotubes. In the work of Diekmann et al. [81] dealing with PDMS/MWCNT composites, it was shown that a pre-dispersion step in a solvent by ultrasonication leads to an increase in electrical conductivity in the composites with regard to composites filled with MWCNTs without pretreatment but deteriorates the mechanical performance of the composites. Physical functionalization by using surfactants did not enhance the electrical and mechanical performance and chemical surface modifications based on oxidation-method-enhanced polymer–filler interactions but lead to tube damage and surface defects. For NR/CNT composites, Nakaramontri et al. [60] used the silane coupling agent “Si69” already applied in the case of silica-filled hydrocarbon rubbers. The surface functionalization of the tubes was carried out by two methods: the CNTs were treated ex situ by acid and “Si69” before mixing with the rubber or the before the silanization process takes place during the composite processing in the internal mixer. It is shown that the in situ method yields better modulus, tensile strength and electrical conductivity but reduces the elongation at break with regard to the unfilled sample. Nakaramontri et al. [82] also showed that modified NR molecules such as epoxidized NR (ENR) or maleated NR (MNR) filled with CNTS display lower electrical percolation threshold as a result of a better dispersion due to interactions between functional groups in ENR or MNR and the polar groups on CNT surfaces. A 100% modulus and tensile strength were also improved for the ENR-CNT and the MNR-CNT composites with regard to NR-CNT composites. The improvement in electrical conductivity obtained by the use of a dual filling (CNTs and conductive CB or CNTs and silver nanoparticles) of NR was reported by Krainoi et al. [83]. This approach of adding CB particles to CNTs in a rubber matrix has been already successfully applied to help the dispersion of carbon nanotubes and improve the mechanical and electrical properties [84]. In a SBR rubber, CNTs are seen to connect CB aggregates, thus forming conducting paths in the polymer matrix (Figure 6). Mixtures of double fillers (pyrogenic silica and sepiolite fibers) have also been shown to impart better mechanical properties to SBR than the use of each single filler on account of synergistic effects arising from filler particles of different morphologies [85]. The effect of multi-scale fillers on the electrical conductivity and resistivity of carbon black/carbon fiber-reinforced polymer nanocomposites was investigated by Haghgoo et al. [86] using a multistep analytical model. A lowering in the percolation threshold and an enhancement in the conductivity up to several orders of magnitude was demonstrated by the introduction of multi-scale fillers into the polymer.

Clay nanofibers of sepiolite have also been used to reinforce elastomeric matrices [44,87,88]. Chemical modification processes of the pristine sepiolite developed and patented by Tolsa make it compatible with low-polarity polymers. Incorporation of the organophilic sepiolite to natural rubber has been shown to impart excellent mechanical properties to the rubbery matrix on account of the needlelike structure of the nanoclay and its ability to align in the direction of strain combined with an interfacial polymer-filler bonding [44]. Interestingly, in the presence of sepiolite, the strain-induced crystallization of natural rubber occurs at a lower strain than that of the unfilled sample, and a higher orientational level of polymer chains has been obtained [89].

## 5. Layered Fillers

Layered solids consisting of two-dimensional layers are good candidates as reinforcing fillers in elastomeric compounds due to their anisotropic shape and their lamellar ability to be exfoliated, thus giving rise to a large polymer–filler interface. Furthermore, plate-like fillers can enhance the gas barrier properties of polymeric matrices which is a particularly important advantage of these materials to produce technical rubber goods with reduced gas permeability [90]. A wide range of layered materials, including phyllosilicates, layered oxides, layered double hydroxides, and graphite, have been studied extensively as promising candidates for the reinforcement of various types of polymer matrices. Layered clays have attracted significant interest as a reinforcing nanofiller for rubbers on account of their low cost, their light weight and their unique layered structure that can be organically modified to make them compatible with hydrophobic polymers. The extent of property improvement of the composite depends on the degree of delamination or exfoliation of the silicate layers. Complete exfoliation is the ideal case for the physical performance of the resulting composite. Most of the recent studies carried out on composites based on natural rubber show that organoclays display exfoliated structures with well dispersed layers in the host matrix and interfacial interactions with the polymer chains [91,92,93,94,95].

The present work only focuses on graphitic materials and especially on graphene, the single layer of sp^2^ hybridized carbon atoms considered as the basic building unit for various carbon allotropes. Its outstanding mechanical, electrical and thermal properties make graphene and its derivatives (graphite oxide, graphene oxide and reduced graphene oxide) the most fascinating materials in the field of nanocomposites [96,97,98,99].

The various ways of preparing samples of single- or few-layer graphene are well documented in the literature. There are a number of reports that describe the synthesis methods based on mechanical or liquid-phase exfoliation, chemical vapor deposition or reduction in exfoliated graphene oxide layers that starts from graphite oxide [90,96,97,98,99,100,101,102]. The graphite oxide route is probably the most promising technique for large-scale production of graphene-based composites. Typically, graphite is submitted to an oxidation process that introduces oxygen-containing functional groups on its surface. The oxidized structure can be easily exfoliated to individual graphene oxide (GO) sheets that can be subsequently reduced to graphene in order to partially restore the electrical conductivity that has been significantly reduced by the structural changes occurring during oxidation. On the other hand, graphene and its derivatives often require a functionalization process in order to improve the polymer–filler interactions and the filler dispersibility [101]. Several examples of surface functionalization of graphene and derivatives have been reported [103,104,105].

Fu et al. [106] prepared natural rubber (NR) composites filled with 1.76 wt% of graphene (GE) or (CNTs) by ultrasonically assisted latex mixing. GE was obtained by oxidation of natural flake graphite to graphene oxide followed by an in situ reduction process. It was shown that at a same filler loading, GE imparts better mechanical properties to the rubber matrix than CNTs. It was also demonstrated that the strain-induced crystallization which is a specific feature of natural rubber characterized by an abrupt increase in stress starts at the lowest strain value in the NR/GE sample with regard to the NR/CNTs and the unfilled rubber. NR/GO nanocomposites with three different size of GO sheets were investigated by Wu et al. [107]. The smaller sheets provide a better reinforcement effect and are more effective in accelerating the strain-induced crystallization of NR. The decrease in strain values at the onset of crystallization which is a common feature of NR composites filled with different types of fillers can be attributed to different factors [62,63]. In the case of highly interacting species as in the case of NR and CB, an increase in the cross-linking density by polymer–filler interactions yield a higher orientation of polymer chains upon stretching, which favors the formation of crystallites. Polymer chains can also be overstrained by strain amplification effects due to the inclusion of undeformable filler particles. The strain amplification phenomenon is noticeable at high filler loadings and in filler-rich areas of the sample [19]. In the case of NR/CNTs or NR/GE where polymer–filler interactions have been shown to be weak [62,63], the occurrence of crystallization at lower strains than that observed in the unfilled rubber is most likely due to the ability of the anisotropic fillers to orientate along the direction of strain, thus affecting the orientation of rubber chains.

She et al. [108] used epoxidized natural rubber (ENR) in order to enhance the interfacial interaction between the GO and the NR matrix via hydrogen bonding between oxygenous functional groups present on the GO surface and the epoxy and hydroxyl groups introduced into the NR chains. The ENR/GO composites were prepared by latex compounding and subsequently vulcanized with static in situ vulcanization and two-roll mixing, respectively. The two vulcanization processes lead to increases in tensile strength and modulus with the addition of GO, but better reinforcement was achieved for composites submitted to static in situ vulcanization. The modulus at a 200% elongation for the sample filled with a 0.7 wt% GO has been found 8.7 times higher than that of the unfilled rubber and only 3.4 times higher with the two-roll mixing.

Silicone rubbers filled with carbon-based nanomaterials have been the subject of several investigations. One of the major advantages of these carbon nanofillers is the ability to provide electrical conduction to the insulating matrix at a very low filler loading [109]. A number of studies reported on the synthesis and characterization of silicone composites filled with graphenic materials promised to be useful in various applications [105,110,111,112,113,114]. Xu et al. [110] used three different processing methods (mechanical mixing, solution blending and ball-milling procedure) for the synthesis of graphene-filled silicone composites. Solution blending yields better electrical conductivity, while better improvement in mechanical properties is obtained with the ball-milling-treated composites. Nevertheless, the results are far below those obtained in MWCNT/silicone rubbers as seen in Figure 7 that compares the dependence of elastic modulus on the filler content of silicone composites filled with graphene and MWCNTs. Outstanding changes have been observed in physical properties of silicone rubbers upon addition of a very tiny amount of carbon nanotubes [109,115]. This surprising affinity of carbon nanotubes for poly(dimethylsiloxane) has been assumed to be ascribed to CH-π interactions between the methyl groups of PDMS and the π-electron-rich surface of carbon nanotubes. This type of interaction could also exist with graphene, but the presence of defects in the structure produced by processing methods may explain the poorer results.

As already mentioned, an interesting approach regarding the reinforcement process of elastomeric materials is the use of hybrid fillers of different morphologies in order to achieve better properties of the composite than those obtained with a single filler. Carbon black [116], carbon nanotubes [117,118] or carbon dots [119] have been added to graphene or graphene oxide to create hybrid structures intended to improve the performance of the host matrix. Paduvilan et al. [89] compared the hybrid effects of CB with two different nanofillers (GO and a nanoclay) in a chlorobutyl–natural-rubber elastomer blend. It was shown that the addition of 0.1 phr of GO to 20 phr of CB drastically decreases the permeability of air while a 3 phr nanoclay loading is required to produce the same decreasing trend in air permeability. In a review of Barshutina et al. [116] devoted to silicone composites with CNT/graphene hybrid fillers, the authors report the existing approaches for the synthesis of hybrid composites with a seamless, assembled and/or foamed structures. Chen et al. [117] showed that partially substituting CB with graphene/CNTs in trans-1,4-polyisoprene/natural rubber composites leads to better mechanical performances. Sreenath et al. [119] also used hybrid nanofillers of GO and carbon dot nanoparticules (CDs) synthesized from citric acid and glycine to prepare nanocomposites based on carboxylated acrylonitrile butadiene latex (XNBR). The amine groups of CDs are covalently linked to the -COOH groups of GO and of the polymer latex which increases the interaction between the nanofillers and the latex. The CDs-GO hybrid nanofiller containing 2 phr of CDs and 2 vol% of GO has been found to yield superior mechanical properties compared to those imparted by pure GO or pure CD-filled XNBR latex.

Table 1 summarizes the characteristics of common fillers used in elastomeric materials and Table 2 compares their effects on elastomer properties.

## 6. Conclusions

The mechanisms of reinforcement of rubbers by conventional fillers highlighted the relevant parameters expected to improve the matrix physical performance. The morphology of the filler particles, their volume fraction and state of dispersion in the host medium as well as their surface characteristics and the chemically active sites that determine the filler–filler and polymer–filler interactions were shown to strongly affect the mechanical properties of the filled elastomers. Poor interfacial interactions between the two phases require modification of the filler surface or the use of a coupling agent able to establish molecular bridges at the polymer–filler interface. The advent of nanoparticles that exhibit high specific area paved the way to the synthesis of advanced composite materials. Used at lower loading content than conventional fillers, these nanoparticles make easier the composite processing and yield lighter final materials.

In situ generation of particles by a sol–gel method was shown to be an effective approach to achieve a homogeneous dispersion of small-sized particles in the host matrix essentially when the process is carried out in the already preformed elastomeric networks. The reinforcing strategy consisting of generating particles after the cross-linking process leads to interpenetrated networks with mechanical properties different from those of conventional composites at relatively high filler loadings. But despite the fact that the sol–gel procedure has an enormous potential for the synthesis of high-performance rubber nanocomposites, the high cost of precursor materials has not enabled huge development of this technique at an industrial scale. On the other hand, the in situ filling process by swelling the rubber after vulcanization seems difficult to handle in the industry.

Among spherical particles, carbon dots are emerging as one of the most exciting nanomaterials in materials science. Owing to their small size, their good dispersibility in host media, their ability to yield strong interfacial interactions, their fluorescence properties and their fabrication from inexpensive synthesis methods from green precursors, they will soon become quite attractive reinforcing nanofillers for polymeric matrices.

The use of anisotropic carbon-based nanomaterials such as carbon nanotubes (CNTs) or graphene derivatives in polymer nanocomposites relies on their high aspect ratio and surface area, thus allowing a large polymer–filler interface. CNTs impart remarkable mechanical and electrical performance to polymeric matrices, essentially attributed to their outstanding intrinsic mechanical properties and their high aspect ratio allowing the formation of a percolated network at a very low filler content. But the results remain below those expected on account of their poor dispersion and their poor adhesion to the matrix. Further improvement in CNT dispersion and interfacial interactions has been obtained by physical or chemical functionalization of the inert tube surface.

Research on graphene and its related materials such as graphene oxide and reduced graphene oxide has reached an enormous interest in the field of nanocomposites on account of the two-dimensional structure of these nanofillers and of their exceptional mechanical, electrical, thermal and gas barrier properties, even better than those of CNTs. But despite their high promise as reinforcing nanomaterials and the development of synthetic methods for the production of single layers, it is still difficult to obtain complete exfoliation, good dispersions in the host matrix and a strong polymer–filler interface. On the other hand, the preparation of single- or few-layer graphene from graphene oxide generates a number of defects that lowers the expected properties. Owing to the high potential of graphene in various applications, further research in this area will definitely overcome the key challenges required for the development of high-performance graphene-based composites.

Finally, it is worth mentioning the growing use of hybrid fillers of different morphologies as a help to assist the dispersion of particles in the polymeric medium on account of synergistic effects that could arise between different filler characteristics.

## Figures and Tables

**Figure 1 polymers-15-02900-f001:**
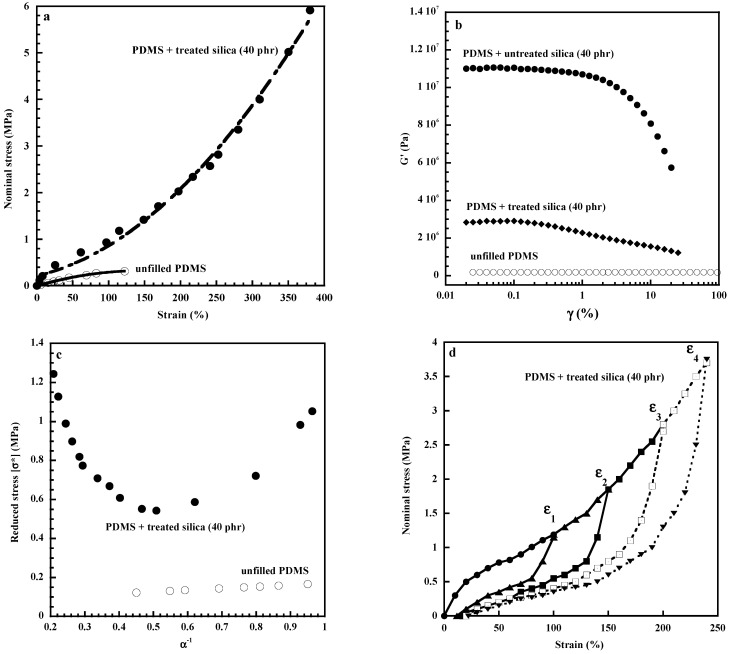
Effect of silica addition on the mechanical response of poly(dimethylsiloxane) (PDMS) networks: (**a**) stress–strain curve of unfilled and filled rubber, (**b**) Payne effect, (**c**) Mooney–Rivlin plots, (**d**) Mullins effect. All the experiments were recorded at room temperature. Materials: Peroxide-cured vinyl-pendant PDMS gum; filler: silica (Aerosil A300 from Degussa), specific area 300 m^2^/g.

**Figure 2 polymers-15-02900-f002:**
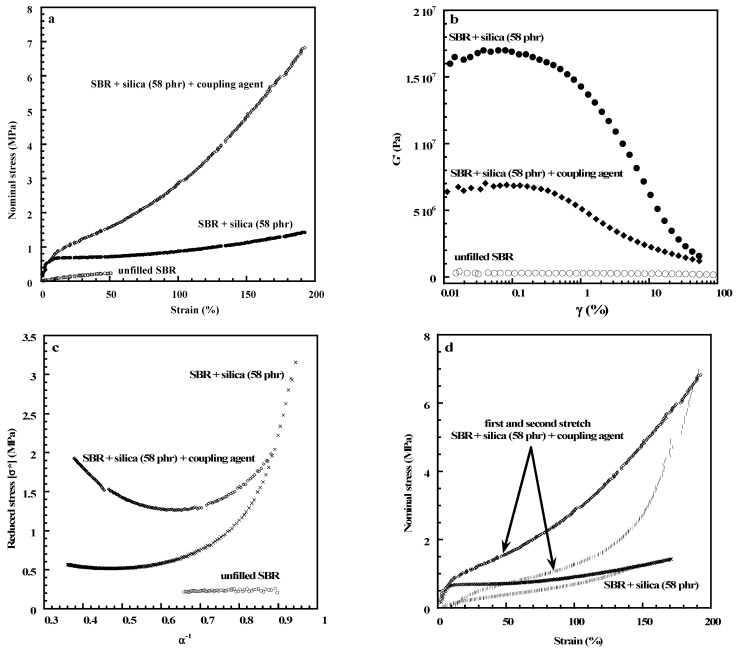
Effect of silica addition on the mechanical response of styrene–butadiene rubber (SBR) networks with and without a coupling agent: (**a**) stress–strain curve of unfilled and filled rubbers, (**b**) Payne effect, (**c**) Mooney–Rivlin plots, (**d**) Mullins effect. All the experiments were recorded at room temperature. Materials: Sulfur-cured SBR (VSL 5525-1 from Bayer, containing 25 w% of styrene units and the microstructure of the butadiene phase is 10% cis, 17% trans and 73% 1,2); filler: highly dispersible silica (Zeosil 1165 MP), specific area 150 m^2^/g.

**Figure 3 polymers-15-02900-f003:**
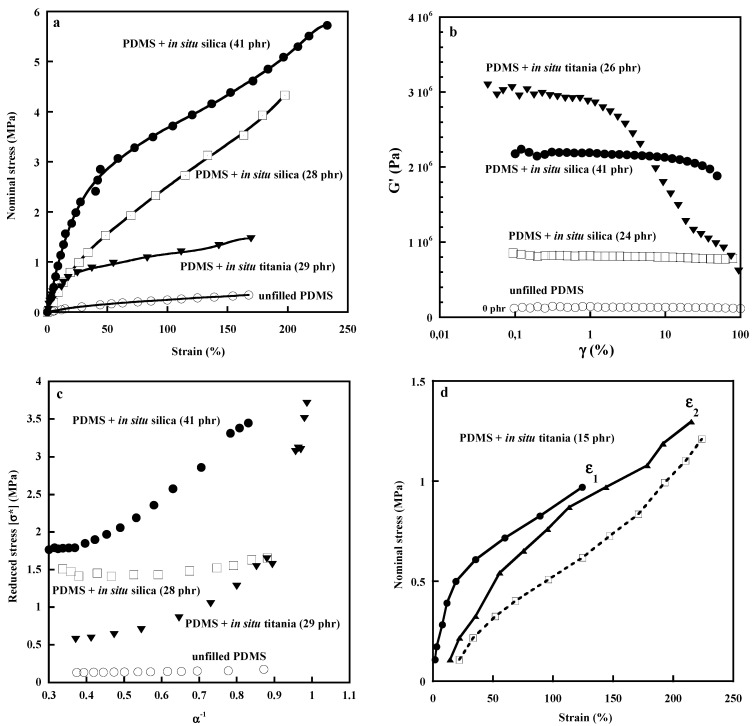
Effect of silica and titania particles generated in situ in PDMS networks: (**a**) stress–strain curve of unfilled and filled rubbers, (**b**) Payne effect, (**c**) Mooney–Rivlin plots, (**d**–**f**) Mullins effects. All the experiments were recorded at room temperature.

**Figure 4 polymers-15-02900-f004:**
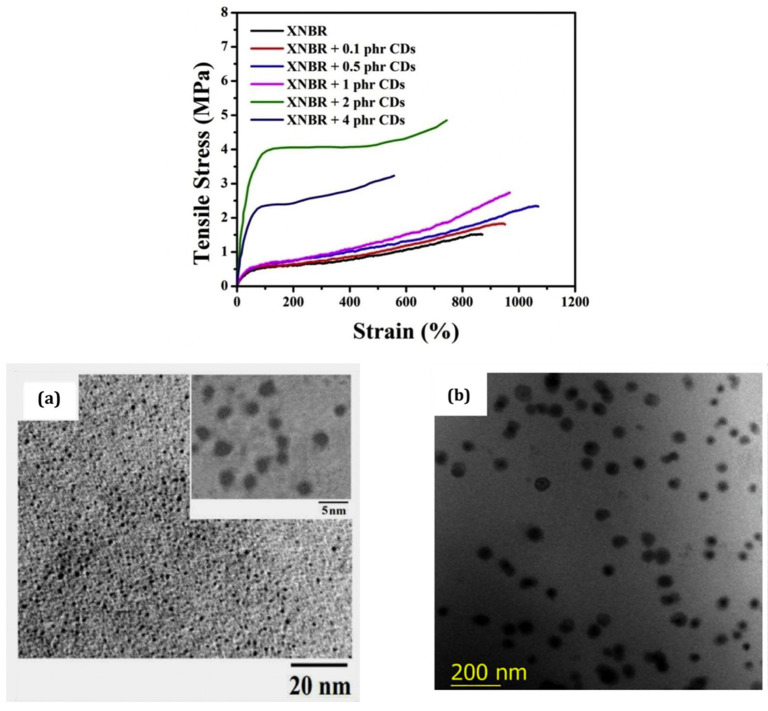
Stress–strain curves of neat carboxylated acrylonitrile butadiene latex (XNBR) and XNBR latex filled with different carbon dots (CDs) contents. (**a**) TEM image of pristine CDs; (**b**) TEM image of XNBR + 2 phr CDs. Source: Reprinted with permission from Ref. [51].

**Figure 5 polymers-15-02900-f005:**
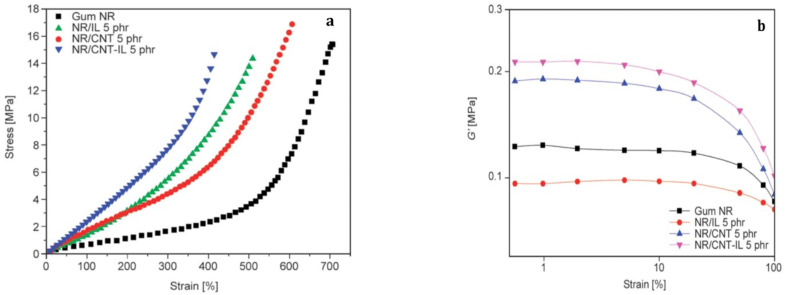
Unfilled natural rubber network (NR) and NR composites filled with carbon nanotubes in the absence (NR/CNT) or presence of a ionic liquid (NR/CNT-IL): stress–strain measurements (**a**); Payne effect (**b**). Source: Reprinted with permission from Ref. [76].

**Figure 6 polymers-15-02900-f006:**
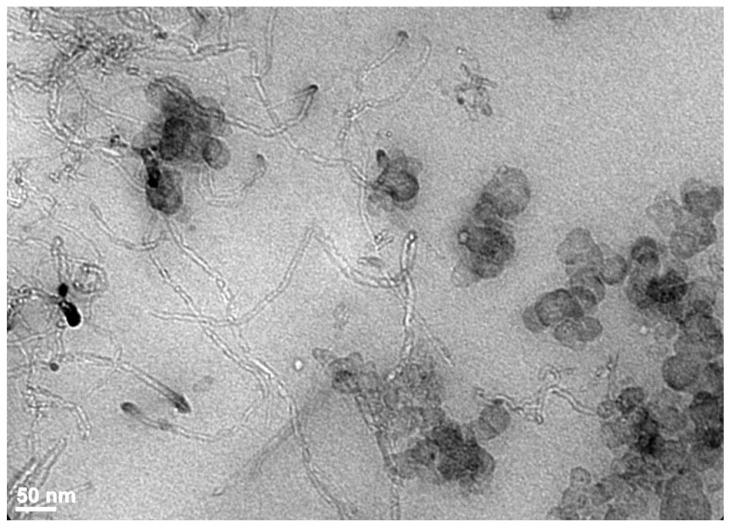
TEM image of styrene–butadiene rubber (SBR) filled with a double filling (5 phr of CB + 5 phr of MWCNTs). Source: Reprinted with permission from Ref. [84].

**Figure 7 polymers-15-02900-f007:**
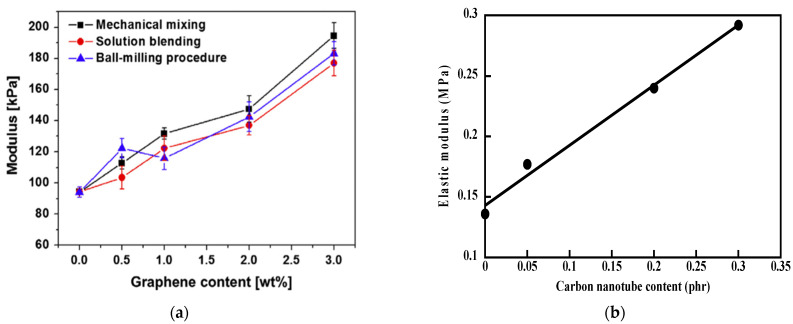
Elastic moduli of silicon rubbers filled with graphene (**a**) and MWCNTs (**b**). Source: Reprinted with permission from Ref. [110] for (**a**).

**Table 1 polymers-15-02900-t001:** Characteristics of common fillers used in elastomeric materials.

Properties	Carbon Black [120,121]	Carbon Nanotubes MWCNTs [122,123]	Graphene [124,125]	Layered Clays Ex: Montmorillonite [126,127]	Silica [3]	Nano Silica [32,33,34,128]	Nano Titania [33,34,42,43]
Dimensions	10–500 nm	Diameter: 10–50 nm Length: micrometers to millimeters	Lateral size greater than 100 nm	Lateral dimensions: 100–1000 nm Layers 1 nm thick	10–100 nm	Diameter: 5–40 nm	Diameter: 20–150 nm
Modulus	~10 MPa	~1 TPa	~1 TPa	178 GPa	73 GPa		
Tensile strength	2–20 MPa	~60 GPa	130 GPa	36 MPa	~50 MPa		
Surface area (m^2^/g)	10–300	>50 ^a^	2630	Up to 800 if complete exfoliation	100–400		
Electrical conductivity (S/cm)	10–10^4^	10^3^–10^5^	~1.5 × 10^4^ (monolayer)				

a: the specific surface area depends on the diameter, the number of walls and the number of nanotubes in a bundle according to ref. [123].

**Table 2 polymers-15-02900-t002:** Effect of various fillers on elastomer properties.

Carbon black	- Still dominant filler in the rubber industry - Improves modulus, tensile strength, abrasion and thermo-oxidative resistance - Large amounts are required to obtain the desired properties which reduces the processability of the materials - Black color is imparted to the finished products
Carbon nanotubes MWCNTs	- Strong mechanical reinforcement on account of their anisotropic character and their ability to orientate in the direction of stretch - Strong reduction in the strain at break due to the presence of bundles that act as failure points - Poor interfacial bonding with the polymer matrix - Electrical percolation threshold around 0.5 phr - Improvement in thermal properties
Graphene	- Very high surface area accessible to polymer chains - Exceptional mechanical, electrical, thermal and barrier properties; conductivity - The properties are extremely dependent on exfoliation
Layered clays Ex: Montmorillonite	- Mechanical reinforcement - Formation of a filler network at low clay content - Improvement in barrier properties - Improvement in thermal stability - Flame retardancy
Silica	- Excellent reinforcing filler for silicone rubbers - Difficult to disperse in non-polar hydrocarbon rubbers due to the formation of a strong filler–filler network, but used with a coupling, it offers several advantages over carbon black in pneumatic manufacture (lower rolling resistance and better wet grip)
Nano silica	- Good optical transparency of the nanocomposites due to the small size of the particles obtained by the sol–gel method - Improved dispersion of the particles within the rubber matrix essentially when the sol–gel process is carried out in the already preformed networks - Improvement in various properties due to the high polymer–filler interface - The silica surface can be chemically modified with silane coupling agents in order to tailor interactions with the polymer matrix - It is difficult to carry out the sol–gel process at an industrial scale on account of the large amounts of precursors required and their cost
Nano titania	- Less improvement in mechanical properties of silicone composites than that imparted by in situ-generated silica particles at a same filler loading due to the larger size of titania particles - Better tensile properties of the composites than those of the externally filled titania at a same filler loading - Further enhancement in the mechanical properties of titania-filled hydrocarbon rubbers with the use of coupling agents - Strong antibacterial properties - Good resistance to UV radiation

## Data Availability

Data available from the author.

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
