# Peer review of "Elastomer Nanocomposites: Effect of Filler–Matrix and Filler–Filler Interactions"

_polymers, 2023, doi:10.3390/polym15132900_

Round 1

Reviewer 1 Report

This manuscript reviews the mechanisms of nano-filled elastomers. The authors categorize nano-fillers into three distinct shapes: spherical, rod-shaped, and layered fillers. However, the relationship between the geometry of nano-fillers and the characteristics of elastomers is not discussed. No discussion on the type of elastomer and the structures. Also, the authors should add a Table summarizing the type of nano-fillers, their shape, and their properties.  Other comments.

1.       References must be included in Figures 1, 2, and 3.

2.  Section 2, Basic mechanism of filler reinforcement in conventional composites. In this section, only silica (a spherical particle) was discussed. Other nano-filler should be discussed as well.

3.       The effect of nano-fillers' size on elastomers' properties should also be discussed.

4.       In section 4, rod-shaped particle. In this section, only CNT were mentioned. Different materials should be discussed.

5.       Table comparing the different types of nano-fillers on elastomer characteristics should be added. 

Author Response

Dear Reviewer 1,

Thank you very much for your comments.

-The geometry of the different nanofillers are discussed in the text: spherical, rod-shaped and layered structures.

--There are no references for Figures 1, 2 and 3 because all the data arise from my own research. I will precise it in the text.

-I described the basic mechanisms by taking the example of silica-filled elastomers but these mechanisms (increase in the elastic modulus, polymer-fillers interactions, Payne and Mullins effects….) are valid for any type of nanofiller. Nevertheless, I have added a recent reference related to the reinforcement imparted by carbon black.

-I have mentioned, as required, references on other nanofillers (fibrous and layered clays).

Thank you very much.

Reviewer 2 Report

This article is devoted to the review of fillers reinforcement mechanism of elastomer matrix. The key points of matrix-filler and filler-filler adhesion mechanisms as well as the impact of fillers type, concentration and shape on the mechanical properties were significantly described. However, this article can be recommended for publication after minor revisions according to the below comments.

The title of the article should be modified because its content involves just matrix&fillers adhesion and mechanical properties information. In contrast, review must be more extensive and touched other crucial aspects. For instance, what is about residual stresses that are arising during elastomer filling, vulcanization and carbonization processes? What is about composite shrinking and relaxation mechanism? What is about toughness and cracks initiation and propagation mechanism for such composites?

Author Response

Dear Reviewer 2,

Thank you very much for comments.

-I completely agree with your comment regarding the title, I have changed it.

-Regarding the questions you mentioned on the residual stresses and others, I have not worked on these aspects. Nevertheless, I brought some answers and references.

I would like to thank you again for your comments.

Reviewer 3 Report

The paper is a review about the mechanisms of reinforcement, but throughout the content, such information is very little. The mechanisms should be further discussed with the related theories and equations.

Besides, the novelty of the review is not emphasized. There are other similar reviews published before, but the author did not mention them clearly in the Introduction section and provide the reasons on why this new review is needed. The knowledge gap is missing and the objectives of the review is NOT clearly written at the end of first section.

Abstract – Objective is not clear and the conclusion is very weak. What are the main outcomes of this review? What the readers can benefit from the review?

Introduction – There is NO any reference cited for this section!

Figure 1, 2 and 3 – Where are the references for each image? Cite it accordingly!

Line 100 – Figure illustrates…….This figure was already mentioned in Line 58.

Figure 3(e) – This image is blur! Besides, all the images’ dimension is NOT consistent!

Figure 4 – The stress-strain curves image is NOT labelled properly! For (a) and (b) – Comparison can only be made by providing the images with SAME scale bar!

Figure 6 – More TEM images should be provided in Section 3, 4 and 5.

A comparison table should be created for different materials (nanospherical particles, rod-shaped particles, layered fillers) by examining their properties and the outcomes. Currently, there is no discussion on which material is best and which one is more practical to be used industrially.

Conclusion is TOO lengthy. Shorten it. Please provide future remarks.

Author Response

Dear Reviewer 3,

Thank you very much for your relevant comments and your thorough review of my paper.

-The mechanisms of reinforcement : increase in the modulus imparted by active fillers, the Payne and the Mullins effects, are all discussed in the section related to the “Basic mechanisms of filler reinforcement in conventional composites” by taking two systems that display a strong and a weak interface. As mentioned in your comment, several reviews have been published in this subject and it is not necessary to rewrite what is already known.

-In view of your comment , I have added my objective in the abstract.

-Several references have bee added in the introduction.

-There are no references for Figures 1, 2 and 3 because all the data arise from my own research. I will precise it in the text.

-Line 100 : I have suppressed Figure 1 illustrates …..

-Figure 3f is blurred: I totally agree with you but I cannot draw it better because I took it from my student’s thesis and I do not have the data to redraw it again. If it is a problem , I can suppress it but it would be a pity because it shows the large residual deformation above the percolation threshold.

-Figure 4 has been reprinted from the paper of . Sreenath, P.R.; Singh, S.; Satyanarayana, M.S.; Das, P.; Kumar in Polymer 2017, 112, 189-200. I cannot change anything.

-It is really difficult to create a table since, except for composites filled with in situ generated particles, the experimental results are far below those expected due to a poor dispersion and weak interface. From my point of view, graphene oxide is the most promising nanofiller for large-scale production of graphene-based composites because it has functional groups that allows surface functionalization and better interfacial bonds with the polymer chains.

I would like to thank you again because it seems to me that taking into account your comments has strengthened the paper.

Reviewer 4 Report

This work reports some results obtained with three types of nanoparticles that  can reinforce 18 rubbery matrices: spherical, rod-shaped and layered fillers. Each type of particle is shown to impart 19 to the host medium a specific reinforcement on account of its own structure and geometry. I believe this paper can be published in the journal if the authors address the following comments effectively and revise their manuscript.
1- The manuscript requires thorough language editing.

2- Quality of some figures is not good.

3- More physical interpretation about the experimental results can improve the quality of this work.

4- A brief discussion about the role of nanoparticle agglomeration in the effective properties of the nanocomposites can improve the quality of this work using these papers: [Composites Science and Technology 209 (2021) 108791] ;[International Journal of Engineering Science 157 (2020) 103392].

The manuscript requires thorough language editing.

Author Response

Dear Reviewer 4,

Thank you very much for your comments.

-Regarding my English, Polymers has a special English revision session after the manuscript is accepted.

-I totally agree with you about the quality of Figure 3f but I cannot draw it better because I took it from my student’s thesis and I do not have the data to redraw it again. If it is a problem , I can suppress it but it would be a pity because it shows the large residual deformation above the percolation threshold.

-I tried to give more physical interpretation by developing the strain-amplification concept which is, from my point of view, of prime importance in the mechanical response of the filled material.

-Thank you for suggesting references. 

Thank you very much.

Round 2

Reviewer 1 Report

I think the authors should add a Table summarizing the type of nano-fillers, shapes, and properties. Also, a Table comparing the effects of various nano-fillers on elastomer properties should be included.

Author Response

Answers to Reviewer 1

Dear Reviewer 1,

-I have added, as recommended, two tables as well as new references related to the various fillers.

Thank you very much.

Liliane Bokobza

Reviewer 3 Report

I'm not satisfied with the response.

Author Response

Answers to Reviewer 3

Dear Reviewer 3,

-I have added, as recommended, two tables as well as new references related to the various fillers.

Thank you very much.

Liliane Bokobza